# A genome-wide CRISPR screen identifies host factors that regulate SARS-CoV-2 entry

Yunkai Zhu[1,6], Fei Feng[1,6], Gaowei Hu[1,6], Yuyan Wang[1,6], Yin Yu[1], Yuanfei Zhu[1], Wei Xu[1], Xia Cai[1], Zhiping Sun[1], Wendong Han[1], Rong Ye[1], Di Qu[1], Qiang Ding[2], Xinxin Huang[3], Hongjun Chen[4], Wei Xu[5], Youhua Xie[1], Qiliang Cai[1✉], Zhenghong Yuan[1✉] & Rong Zhang[1✉]

The global spread of SARS-CoV-2 is posing major public health challenges. One feature of SARS-CoV-2 spike protein is the insertion of multi-basic residues at the S1/S2 subunit cleavage site. Here, we find that the virus with intact spike (Sfull) preferentially enters cells via fusion at the plasma membrane, whereas a clone (Sdel) with deletion disrupting the multi-basic S1/S2 site utilizes an endosomal entry pathway. Using Sdel as model, we perform a genome-wide CRISPR screen and identify several endosomal entry-specific regulators. Experimental validation of hits from the CRISPR screen shows that host factors regulating the surface expression of angiotensin-converting enzyme 2 (ACE2) affect entry of Sfull virus. Animal-to-animal transmission with the Sdel virus is reduced compared to Sfull in the hamster model. These findings highlight the critical role of the S1/S2 boundary of SARS-CoV-2 spike protein in modulating virus entry and transmission and provide insights into entry of coronaviruses.

[1] Key Laboratory of Medical Molecular Virology (MOE/NHC/CAMS), School of Basic Medical Sciences, Shanghai Medical College, Biosafety Level 3 Laboratory, Shanghai Institute of Infectious Disease and Biosecurity, Fudan University, Shanghai, China. [2] Center for Infectious Disease Research, School of Medicine, Tsinghua University, Beijing, China. [3] Technical Center For Animal, Plant and Food Inspection and Quarantine of Shanghai Customs, Shanghai, China. [4] Shanghai Veterinary Research Institute, CAAS, Shanghai, China. [5] Guangdong Provincial Key Laboratory of New Drug Screening, School of Pharmaceutical Sciences, Southern Medical University, Guangzhou, China. [6] These authors contributed equally: Yunkai Zhu, Fei Feng, Gaowei Hu, Yuyan Wang.
✉email: qiliang@fudan.edu.cn; zhyuan@shmu.edu.cn; rong_zhang@fudan.edu.cn

SARS-CoV-2 and SARS-CoV share nearly 80% nucleotide sequence identity and use the same cellular receptor, angiotensin-converting enzyme 2 (ACE2), to enter target cells[1,2]. However, the newly emerged SARS-CoV-2 exhibits greater transmissibility[3–5]. The viral structural protein, spike (S), plays critical roles in determining the entry events, host tropism, pathogenicity, and transmissibility. One significant difference between the SARS-CoV-2 spike protein and those of other bat-like SARS-CoV is the insertion of multi-basic residues (RRAR) at the junction of S1 and S2 cleavage site[6]. Previous studies showed that expression of SARS-CoV-2 spike in cells promotes cell–cell membrane fusion, which is reduced after deletion of the RRAR sequence or when expressing SARS-CoV S protein lacking these residues[7,8]. Pseudovirus or live virus bearing SARS-CoV-2 spike deletion at the S1/S2 junction decreased the infection in Calu-3 cells and attenuated infection in hamsters[7,9]. The sequence at the S1/S2 boundary seems to be unstable, as deletion variants are observed both in cell culture and in patient samples[9–12]. SARS-CoV-2 entry is mediated by sequential cleavage at the S1/S2 junction site and additional downstream S2′ site of spike protein. The sequence at the S1/S2 boundary contains a cleavage site for the furin protease, which could preactivate the S protein for membrane fusion and potentially reduce the dependence of SARS-CoV-2 on plasma membrane proteases, such as transmembrane serine protease 2 (TMPRSS2), to enable efficient cell entry[13].

Coronaviruses enter cells through two pathways: fusion at the plasma membrane (early pathway) or in the endosome (late pathway) in a cell-type-dependent manner[14]. The presence of exogenous and membrane bound proteases, such as trypsin and TMPRSS2, triggers the early fusion pathway. Otherwise, it will be endocytosed through the late pathway[14]. The low-pH environment in the endosome activates the enzyme cathepsin L (CTSL) to trigger the fusion of virion membrane with endosome membrane to release the genome. In Calu-3 cells that express the protease TMPRSS2 but low CTSL, SARS-CoV prefers to enter through the early pathway[15]. In this study, we isolated a clone of SARS-CoV-2 that has the deletion disrupting the multi-basic residues at the S1/S2 cleavage site in spike protein. This virus preferentially utilized the endosomal entry pathway in A549 cells expressing the receptor ACE2, which provide ideal virus and cell models to dissect the entry. With these models, we performed a genome-wide CRISPR/Cas9 based screen, and identified a suite of host genes that regulate the endosomal entry or surface expression of receptor ACE2 protein. We also determined the impact of endosomal entry-specific virus on the pathogenesis and transmission in a hamster model.

## Results

**The deletion at the S1/S2 boundary of spike protein propels the virus to enter cells through the endosomal pathway.** We observed the same phenomena that others have reported, an instability of the SARS-CoV-2 S1/S2 boundary[9–11]. Using the patient-isolated SARS-CoV-2 SH01 strain, we performed three rounds of plaque purification in Vero E6 cells in the presence of trypsin and observed no mutations in any of the structural genes (Sfull virus). However, after two additional rounds of passage without trypsin, a 21-nucleotide deletion at the S1/S2 cleavage site was acquired, disrupting the RRAR motif (Fig. 1a and Supplementary Fig. 1a). Unexpectedly, this presumed cell culture adaptation could be prevented by adding trypsin to the media or by ectopically expressing the serine protease TMPRSS2 in Vero E6 cells during a continuous passaging experiment (Supplementary Fig. 1b, c). We designated the plaque-purified deletion clone as Sdel virus and detected no additional mutations in the full-length genome when compared to the Sfull virus (Supplementary Fig. 1d). Compared to Sfull, the deletion-bearing Sdel virus

exhibited a dramatic increase in infectivity as measured by the greater percentage of nucleocapsid (N) antigen-positive cells in wild-type Vero E6 (hereafter Vero cells), Vero plus trypsin, Vero expressing TMPRSS2, and A549 cells expressing the receptor ACE2 (Fig. 1b and Supplementary Fig. 2a). Conversely, in human Calu-3 lung epithelial cells, the Sdel virus replicated slower than the Sfull clone (Fig. 1b), similar to previous reports using a pseudovirus or fully infectious, mutant virus[7,9]. Moreover, we found that pseudovirus bearing the S protein from Sfull, Sdel, or a RRAR mutant variant (R682S, R685S)[16] had a phenotype similar to infectious viruses used in these cell types (Fig. 1c). Of note, infection using either the Sdel S- or mutant variant S (R682S, R685S)-bearing pseudovirus was decreased by approximately ten-fold in Calu-3 cells, highlighting the critical role of these basic residues at the S1/S2 boundary in infectivity.

To assess the impact of the S1/S2 junction deletion on viral entry pathways, cells were treated with camostat mesilate, a TMPRSS2 inhibitor that blocks viral fusion at the plasma membrane, and/or E-64d (aloxistatin), an inhibitor that blocks the protease activity of cathepsins B and L, which are required for the endosomal membrane fusion (Fig. 1d and Supplementary Fig. 2b). We observed apparent S1/S2 cleavage for Sfull virus but not for Sdel in multiple cell types (Supplementary Fig. 3a). Sfull virus infection, as measured by N antigen-positive cells, was sensitive to inhibition by E-64d but not camostat in Vero cells (Fig. 1d), suggesting that Sfull virus enters the TMPRSS2-negative Vero cells through the endosomal pathway. When TMPRSS2 was expressed, both camostat and E-64d inhibited the infectivity of Sfull, indicating that expression of TMPRSS2 could promote the membrane fusion entry pathway. Remarkably, E-64d and camostat had no effect on Sfull virus in A549-ACE2 cells, suggesting that in this cell Sfull may use other TMPRSS2 homologs or trypsin-like proteases to activate fusion at the plasma membrane since TMPRSS2 expression is absent in A549 cells[17,18]. We observed a similar phenotype even when cells were treated with a high concentration of inhibitors (Supplementary Fig. 2c). In Calu-3 cells that express the TMPRSS2, camostat completely blocked the Sfull infection, but E-64d had minimal effects, suggesting that Sfull preferentially enters Calu-3 cells via the plasma membrane fusion pathway.

For the Sdel virus, E-64d significantly inhibited infection in Vero, Vero-TMPRSS2, and A549-ACE2 cells, whereas camostat did not reduce the infection, even in Vero-TMPRSS2 cells (Fig. 1d). It is noteworthy that Sdel was sensitive to both inhibitors in Calu-3 cells unlike the Sfull virus, and these two compounds exerted a synergetic effect on Sdel infection. This suggests Sdel utilizes both plasma membrane and endosomal fusion pathways in Calu-3 cells. Notably, the spike protein of SARS-CoV does not have the insertion of multiple basic residues at the S1/S2 cleavage site (Fig. 1a). We found that, E-64, but not camostat, efficiently inhibited SARS-CoV pseudovirus infection in multiple cell types (Supplementary Fig. 2d). In all, these results demonstrate that: (a) the Sfull virus with intact spike protein are endocytosed into the TMPRSS-negative cells (e.g., Vero), (b) the deletion at the S1/S2 junction site propels the virus to enter cells through the endosomal pathway (e.g., A549 and Calu-3), (c) the more efficient endosomal entry is acquired by the deletion at the S1/S2 site (e.g., Vero and A549), and (d) the entry pathways of SARS-CoV may resemble the Sdel virus. So far, we fully characterized the ensodomal entry properties of Sdel virus, which could be served as a model to dissect the entry process in specific cell types.

**Genome-wide CRISPR/Cas9 screen identifies entry factors using Sdel virus as model.** Genome-wide CRISPR/Cas9 screens have enabled the identification of host factors required for

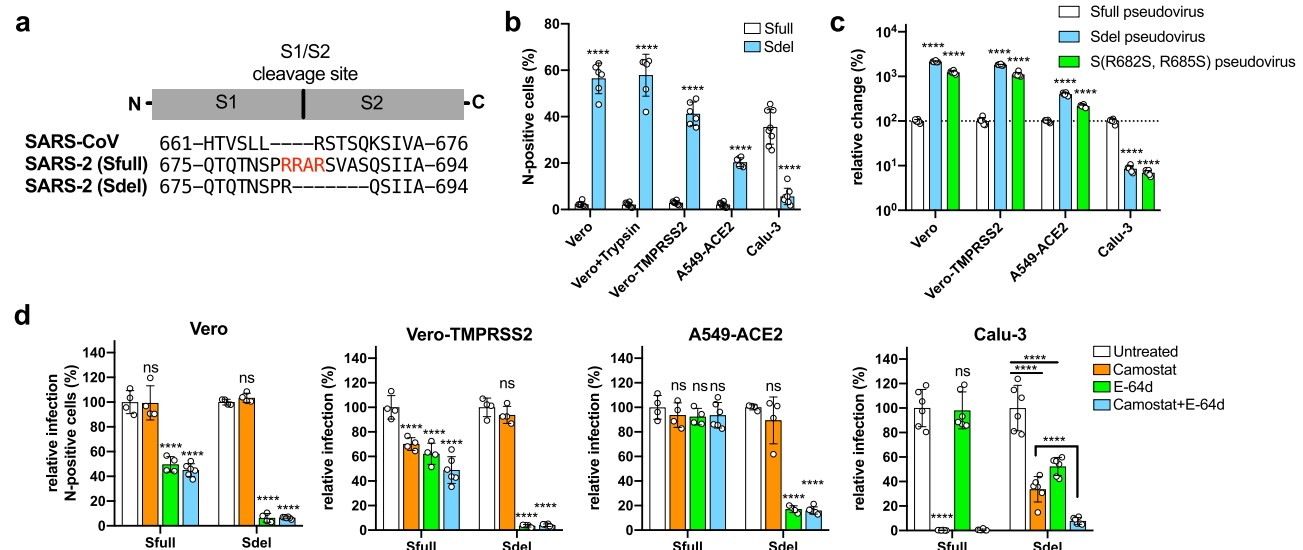

**Fig. 1 The deletion at the S1/S2 boundary of spike protein propels the virus to enter cells through the endosomal pathway. a** Sequence alignment of spike protein encompassing the cleavage site between S1 and S2 subunits. The spike proteins of SARS-CoV-2 without (Sfull strain) and with (Sdel strain) deletion were used to compare with that of SARS-CoV. The insertion of multi-basic amino acids in spike protein of SARS-CoV-2 was shown in red. **b** Comparison of the replication property between Sfull and Sdel strains in different cell lines. The percentage of nucleocapsid (N) protein-positive cells was analyzed by imaging-based analysis following virus infection (two or more experiments; $n = 6$ except for Calu-3 in which $n = 8$; one-way ANOVA with Dunnett's test; mean ± s.d.). **c** Evaluation of entry efficiency in different cell lines infected with pseudoviruses bearing spike protein Sfull, Sdel, or S mutant (R682S, R685S). Data are normalized to the Sfull of individual experiments (two experiments; $n = 6$; one-way ANOVA with Dunnett's test; mean ± s.d.). **d** Effect of TMPRSS2 serine protease inhibitor Camostat and cysteine protease inhibitor E-64d on Sfull or Sdel infection in different cell lines (two experiments; $n = 4$ or 6; one-way ANOVA with Dunnett's test; mean ± s.d.). Data shown were normalized to the untreated group of individual experiments. ****$P < 0.0001$; n.s. not significant.

efficient virus infection[19–23]. A lack of suitable human physiologically relevant cell lines and the spike protein-induced syncytia formation in cells have made such a screen to identify the host factors that regulate the entry of SARS-CoV-2 very challenging. We found that Sdel virus preferentially enters A549-ACE2 cells via the endosomal pathway, replicates robustly, does not cause syncytia, and efficiently results in cell death. Because of these properties, the Sdel virus represented an ideal virus model to investigate the endosomal entry process. To this end, we performed a genome-wide, cell survival-based screen with the Sdel virus in A549-ACE2 cells transduced with a library of single-guide RNAs (sgRNAs) targeting 19,114 human genes (Fig. 2a)[24]. The vast majority of transduced cells inoculated with Sdel virus died within 7 days of infection. Surviving cells were harvested and expanded for a second round of challenge with Sdel. The remaining surviving cells were expanded and subjected to genomic DNA extraction, sgRNA sequencing, and data analysis (Supplementary Data 1).

The top candidates from the CRISPR screen were determined according to their MAGeCK score (Fig. 2b). The top hit was ACE2, the cellular receptor that confers susceptibility to SARS-CoV-2, which confirmed the validity of the screen. Additionally, the gene encoding cathepsin L (CTSL), a target of our earlier assay using E-64d that is known to be important for activating SARS-CoV virion membrane fusion with the endosome[25], also was identified, again confirming the utility of the screening strategy.

We chose the top 32 genes with a cutoff of false discovery rate (FDR) <0.15 to validate. For each specific gene target, A549-ACE2 cells were transduced with two independent sgRNAs and then infected with Sdel virus. The percentage of N protein-positive cells was determined by image-based analysis. Remarkably, editing of all 32 genes resulted in a statistically significant reduction of Sdel infection compared to cells receiving the

control sgRNA (Fig. 2c). Most of these genes were associated with the endolysosome, including components of the retromer complex, the COMMD/CCDC22/CCDC93 (CCC) complex, Wiskott–Aldrich syndrome protein and SCAR homologue (WASH) complex, and actin-related protein 2/3 (Arp2/3) complex, which have significant roles in endosomal cargo sorting[26,27]. We also identified genes encoding the WD Repeat Domain 81 (WDR81)–WDR91 complex, which was detected in a previous genetic screen for regulators of endocytosis and the fusion of endolysosomal compartments[28]. Similarly, we identified the gene encoding Transcription Factor Binding To IGHM Enhancer 3 (TFE3), which may regulate lysosomal positioning in response to starvation or cholesterol-induced lysosomal stress[29]. We also validated NPC Intracellular Cholesterol Transporter 1 (NPC1) and NPC2, which regulate intracellular cholesterol trafficking, as important for Sdel infection[30,31]. In addition, the gene for Activating Signal Cointegrator complex 3 (ASCC3), which functions as a negative regulator of the host defense response, was identified in our screen[32]. From these hits, we selected representative genes to validate for cell-type specificity in HeLa-ACE2 cells, finding that all the genes tested greatly reduced infection with Sdel virus (Supplementary Fig. 4a).

To define the stage of viral infection that each of the 32 validated genes acted, one representative sgRNA per gene was selected for study in A549-ACE2 cells. Due to its known antiviral activity, ASCC3 was not targeted. We confirmed that editing of these genes did not affect cell viability (Supplementary Fig. 4b). The gene-edited cells were infected with pseudovirus bearing the spike protein of Sdel virus or, as a control, the glycoprotein of vesicular stomatitis virus (VSV-G) (Fig. 3a, b). Consistent with data from the fully infectious Sdel virus, editing any of the selected genes markedly inhibited Sdel pseudovirus infection whereas only editing of some retromer-associated genes and the Arp2/3 complex significantly reduced the VSV-G

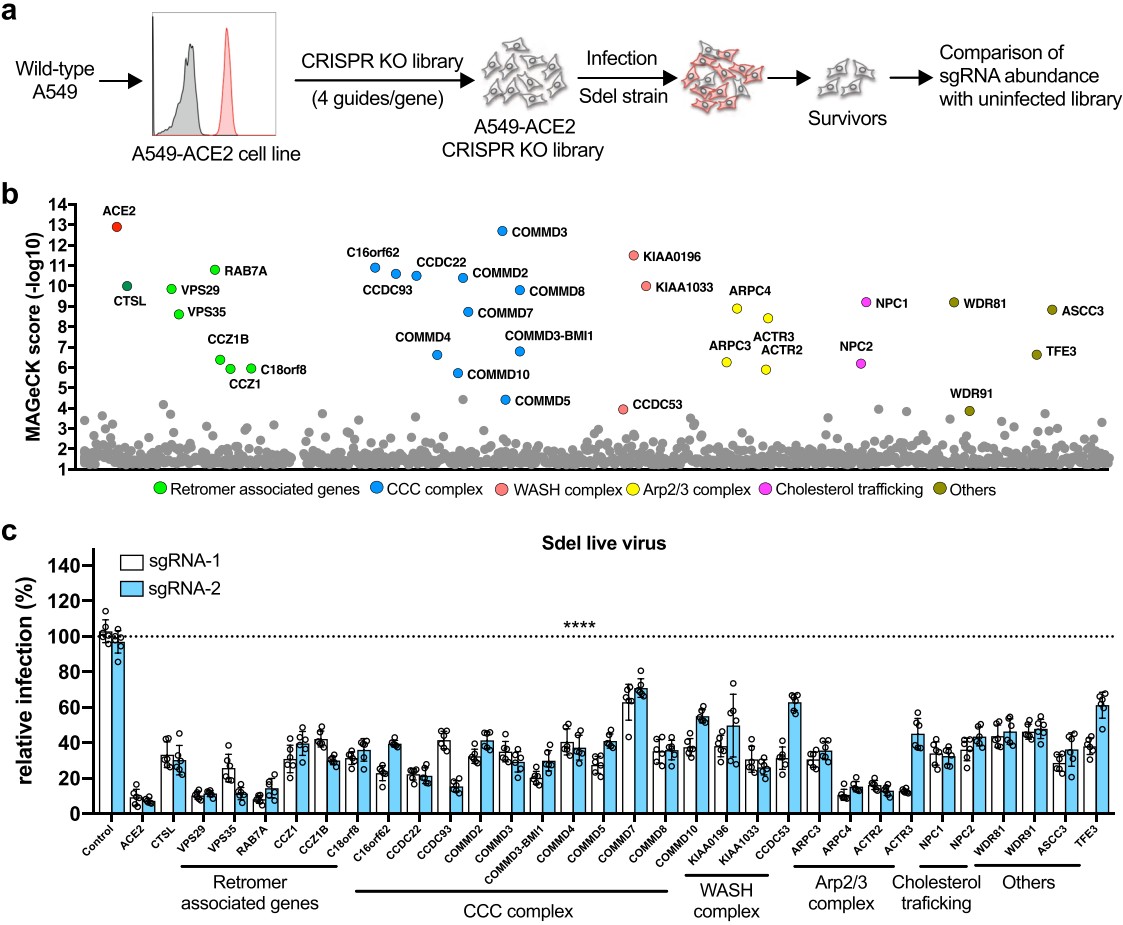

**Fig. 2 Genome-wide CRISPR/Cas9 screen identifies host factors using Sdel virus as model. a** Schematic of the screening process. A549 cells expressing the human ACE2 were used to generate the CRISPR sgRNA knockout cell library. The library was infected with Sdel strain of SARS-CoV-2, and cells survived were harvested for genomic extraction and sequence analysis. **b** Genes and complexes identified from the CRISPR screen. The top 32 (FDR < 0.15) genes were indicated based the MAGeCK score. **c** The top 32 genes were selected for experimental validation in A549-ACE2 cells using two independent sgRNAs by Sdel live virus infection. Data shown are an average of two independent experiments performed in triplicate and are normalized to the controls of individual experiments. One-way ANOVA with Dunnett's test; $n = 6$; mean ± s.d.; ****$P < 0.0001$.

pseudovirus infection (Fig. 3a, b). These results suggest that most of the genes identified act at the virus entry step and are specific to SARS-CoV-2. Notably, pseudovirus bearing the spike protein of SARS-CoV, which lacks the multiple basic residues at the S1/S2 junction as Sdel, exhibited a phenotype similar to Sdel pseudovirus and Sdel live virus (Fig. 3c). Editing of these genes, including those encoding CTSL, cholesterol transporters NPC1/2, WDR81/91, and TFE3, markedly reduced infection, suggesting that Sdel and SARS-CoV may utilize similar entry machinery in this cell type (Fig. 3c). Intriguingly, these edited genes also significantly reduced the infection by pseudovirus bearing the spike protein of Middle East Respiratory Syndrome coronavirus (MERS-CoV) in A549-ACE2-DPP4 cells (Fig. 3d). Although the furin cleavage site is present at the S1/S2 boundary of MERS-CoV[33], it preferentially enters the A549 cell via endosomal pathway as indicated by its sensitivity to E-64d inhibitor (Supplementary Fig. 2e). This is possibly due to the lack of proper protease to activate the plasma membrane fusion pathway in A549 cells for MERS-CoV. In sum, these results indicate that the host genes identified from CRISPR screen are required for the entry in A549 cells by pseudotyped SARS-CoV and MERS-CoV, and SARS-CoV-2 with deletion at the S1/S2 boundary.

To determine whether these genes identified impact Sfull virus infection, one representative sgRNA per gene was tested (Fig. 3e).

Given the low infection efficiency with pseudotyped virions bearing the spike protein of Sfull virus, we only used the Sfull live virus to validate these genes. The editing efficiency of some of these genes by sgRNAs was confirmed by western blotting (Supplementary Fig. 3b). As expected, editing of *CTSL* did not reduce infection (Fig. 3e), as the Sfull virus enters A549-ACE2 cells via an endosomal-independent pathway (demonstrated in Fig. 1d). Also, editing of *NPC1*, *NPC2*, or *TFE3* that functions in endolysosomes had a negligible impact on Sfull virus infection (Fig. 3e). In general, editing of genes encoding complexes that regulate the retrieval and recycling of cargo significantly reduced infection, albeit to a lesser extent than observed with the Sdel live virus (Fig. 3e). U18666A, a cationic sterol, binds to the NPC1 protein to inhibit cholesterol export from the lysosome, resulting in impaired endosome trafficking, late endosome/lysosome membrane fusion[34–36]. U18666A has been shown to inhibit the S protein-driven entry of SARS-CoV, MERS-CoV, and the human coronaviruses NL63 and 229E, with the most efficient inhibition observed with SARS-CoV[37]. The antiviral effect of U18666A on type I feline coronavirus has also been characterized in vitro and in vivo[38,39]. We found that, pretreating A549-ACE2 cells 2 h prior to or post infection had no inhibitory effect on Sfull virus (Fig. 3f). In contrast, Sdel virus was more sensitive to U18666A, even when used for treatment 2 h post infection, presumably due to Sdel preferential usage of the endosomal entry

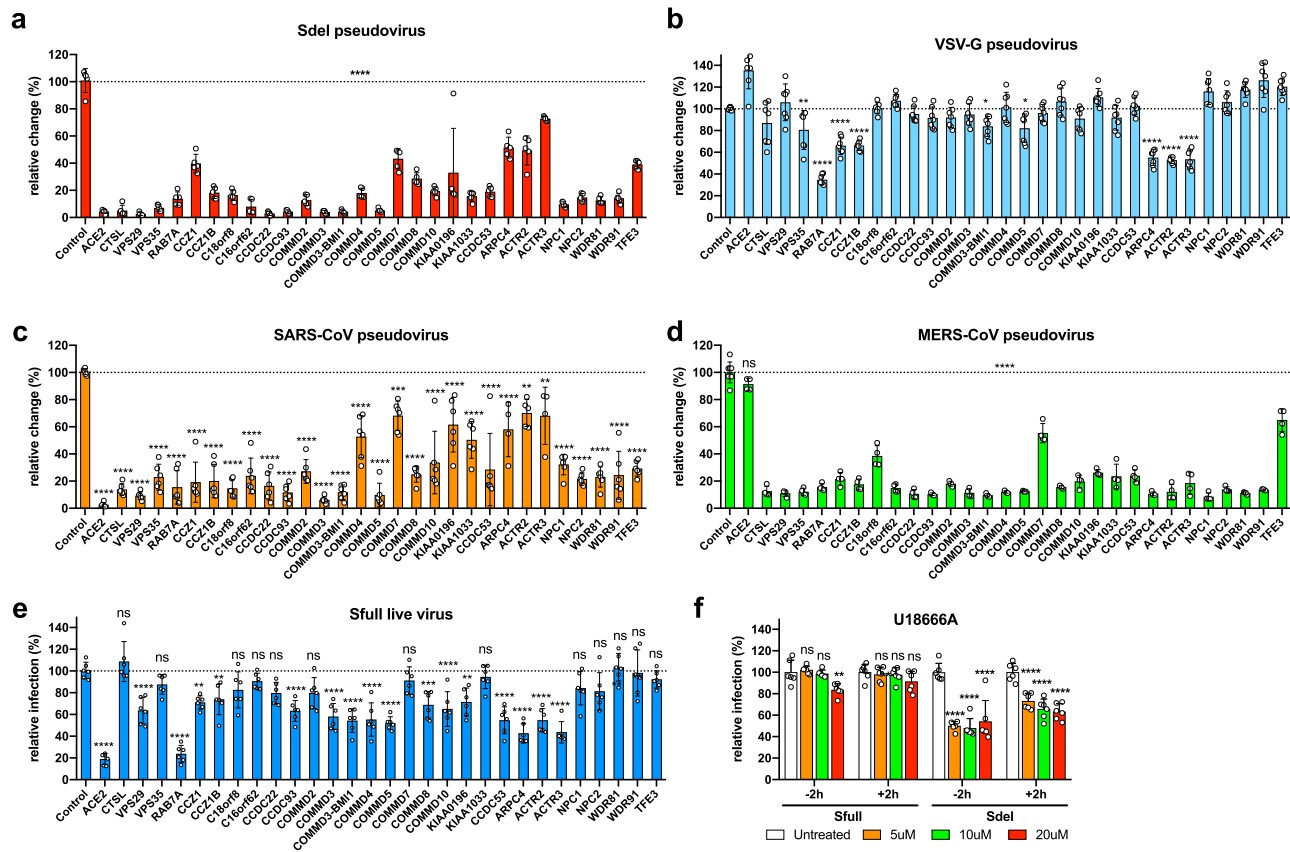

**Fig. 3 Genes identified are required for the endosomal cell entry of SARS-CoV-2, SARS-CoV, and MERS-CoV. a–d** The selected genes were verified for the infection by pseudovirus bearing the spike protein of SARS-CoV-2 Sdel strain (**a**), the the glycoprotein of vesicular stomatitis virus (VSV-G) (**b**), the spike protein of SARS-CoV (**c**), or the spike protein of MERS-CoV (**d**) (two experiments; $n = 4–11$; one-way ANOVA with Dunnett's test; mean ± s.d.). One representative sgRNA per gene was used in A549-ACE2 cells. **e** The genes selected were verified for the infection by the SARS-CoV-2 Sfull live virus (two experiments; $n = 6$; one-way ANOVA with Dunnett's test; mean ± s.d.). **f** Effect of NPC1 inhibitor U18666A on virus infection. Cells were treated with U18666A at the indicated concentrations 2 h prior to or 2 h post infection by Sfull or Sdel live virus. The viral N-positive cells were calculated (two experiments; $n = 6$; one-way ANOVA with Dunnett's test; mean ± s.d.). Data shown were normalized to the controls of individual experiments. $**P < 0.01$; $***P < 0.001$; $****P < 0.0001$; n.s. not significant.

pathway in A549 cells, which is consistent with the result of its sensitivity to E-64d as demonstrated in Fig. 1d.

**Host genes that regulate the surface expression of receptor ACE2 are identified.** The Sdel-validated genes that also affected Sfull infectivity were largely multi-protein complexes (Figs. 2c and 3e). These complexes are important for maintaining plasma membrane and lysosomal homeostasis by maintaining expression of key integral proteins, including signaling receptors and transporters[26,40]. We hypothesized that disruption of these complexes might affect the binding or transit of virions. To this end, we performed binding and internalization assays using Sfull virus in A549-ACE2 cells. The genes COMMD3, VPS29, and CCDC53, which encode proteins that comprise CCC, retromer, and WASH complexes, respectively, were each edited; effects on expression were confirmed by western blotting except for the VPS29 (Supplementary Fig. 3b). Notably, binding and internalization of Sfull virions to these gene-edited bulk cells was significantly decreased when compared to the cells edited with control sgRNA (Fig. 4a).

The entry receptor ACE2 is critical for SARS-CoV-2 infection. To determine whether cell surface expression of ACE2 is regulated by these complexes, gene-edited cells (COMMD3, VPS29, VPS35, CCDC53, CCDC22, and NPC1) were incubated with S1-Fc recombinant protein or an anti-ACE2 antibody, and

binding was measured by flow cytometry (Fig. 4b, c). It showed that editing of these genes apparently perturbed the surface expression of ACE2, with the exception of the cholesterol transporter gene NPC1. To further confirm these findings, we biotinylated the surface proteins of these gene-edited cells, immunoprecipitated with streptavidin, and performed western blotting and quantification (Fig. 4d, e). A significant reduction of surface ACE2 was observed across the different gene-edited cells except for NPC1-edited cells, consistent with the data above from flow cytometry-based surface binding assays. To correlate the significance of this finding for virus infection, we edited CCDC53, which had the greatest reduction in virion internalization as shown in Fig. 4a, in Calu-3 lung cells. Viral yield was approximately ten-fold lower in the CCDC53-edited bulk cells compared to control sgRNA-edited cells at 24 h for Sfull and 48 h for Sdel (Fig. 4f, g). These results suggest retrieval and recycling complexes identified in our screen regulate expression of the ACE2 receptor, which is required for optimal SARS-CoV-2 infection.

**SARS-CoV-2 entry is elegantly regulated by endosomal cargo sorting complexes.** To distinguish the complexes important for virus infection, we edited additional genes. The retriever complex is another retromer-like complex that mediates cargo recycling and consists of the genes DSCR3, C16orf62, and VPS29[41].

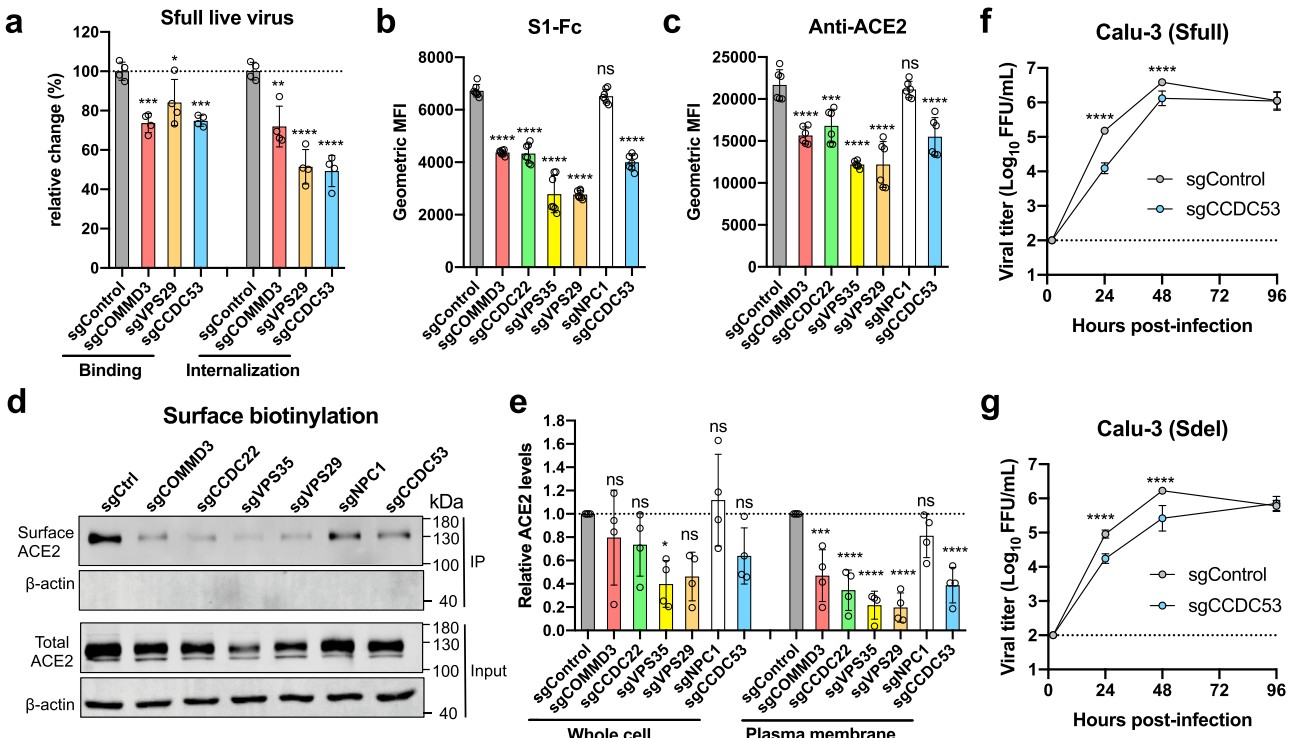

**Fig. 4 Host genes that regulate the surface expression of receptor ACE2 are identified. a** The effect on virion binding and internalization in gene-edited cells. A549-ACE2 cells were incubated with SARS-CoV-2 Sfull infectious virus on ice for binding or then switched to 37 °C for internalization. Viral RNA was extracted for RT-qPCR analysis (two experiments; n = 4; one-way ANOVA with Dunnett's test; mean ± s.d.). **b, c** Surface expression of receptor ACE2 was decreased in gene-edited cells as measured by flow cytometry using S1-Fc recombinant protein or anti-ACE2 antibody (2 experiments; n = 7 (**b**) or 6 (**c**); one-way ANOVA with Dunnett's test; mean ± s.d.). **d, e** Surface and total expression of receptor ACE2 were decreased in gene-edited cells. The plasma membrane proteins were biotin-labeled and immunoprecipitated by streptavidin beads for western blotting. One representative blot was shown (**d**) and data are pooled from four independent experiments, quantified, and normalized to the controls of individual experiments (**e**) (four experiments; n = 4; one-way ANOVA with Dunnett's test; mean ± s.d.). **f, g** The impact on viral production in *CCDC53* gene-edited Calu-3 cells. The mixed cell population was infected with Sfull (**f**) or Sdel (**g**) to assess the virus yield (two experiments; n = 6; two-way ANOVA with Sidak's test). *P < 0.05; **P < 0.01; ***P < 0.001; ****P < 0.0001; n.s. not significant.

*VPS29* and *C16orf16* that were identified in our screen, also are shared functionally by the retromer and CCC complexes[42,43]. Sorting Nexin 17 (SNX17) acts as a cargo adaptor associated with retriever and the adaptor SNX31 (ref. [41]). SNX27 and SNX3 are two additional cargo adaptors associated with the retromer complex[44]. To test these genes, which were not identified in our screen, we introduced three sgRNAs per gene in A549-ACE2 cells and infected with Sdel virus. The editing efficiency of SNX17 and SNX27 was confirmed by western blotting (Supplementary Fig. 5). Among the genes chosen for examination, only the retromer-associated adaptor SNX27 was required (Supplementary Fig. 5), highlighting the importance of the retromer complex over the retriever one for virus infection.

The COMMD proteins of CCC complex are a 10-member family (COMMD1-10)[45] that act as cargo-binding adaptors[46,47]. Of these 10 proteins, we identified the genes encoding all of them in our screen except for COMMD1, 6, and 9 (Fig. 2c). Knockout of the COMMD1, 6, and 9 increases the low-density lipoprotein cholesterol levels in the plasma membrane, thereby maintaining lipid raft composition[48]. In our experiments, editing each of these three genes as well as cholesterol uptake-related genes (*LDLR*, *SRB1*, *CD36*, *LRP1*) did not impact Sdel infection in A549-ACE2 or HeLa-ACE2 cells (Supplementary Fig. 6a, b), suggesting that these members of the COMMD protein family function differently. Notably, knockout of *COMMD1* did not affect expression of *COMMD3* or *CCDC22* in our study as opposed to previous work (Supplementary Fig. 6c)[46,48], which needs to be

further characterized. Overall, our experiments demonstrate that SARS-CoV-2 entry is regulated by endosomal cargo sorting complexes. Understanding how these complexes regulate the sorting of incoming virions might enable development of host-directed antiviral agents to control COVID-19.

**The switch of virion entry pathways modulates the infection and transmission in hamsters.** In the culture of A549 lung epithelial cells, we demonstrated that the deletion at the S1/S2 boundary of spike protein of SARS-CoV-2 resulted in a switch from the plasma membrane to endosomal fusion pathway for entry. Using this model, we uncovered a suite of host genes that regulate the virion endosomal entry and surface expression of receptor ACE2. In Calu-3 lung cells, which model more physiologically relevant airway epithelial cells, this switch led to a less efficient entry process. Since virus entry is the first step in establishing infection, we hypothesized that deletion at the S1/S2 boundary propelling the viral entry to endosomal pathway might reduce virus infectivity and transmissibility in vivo.

Indeed, using the golden Syrian hamster model, a previous study showed that a SARS-CoV-2 variant with a 30-nucleotide deletion at the S1/S2 junction caused milder disease and less viral infection in the trachea and lungs compared to a virus lacking the deletion[9]. Here, we extended the study and systemically evaluated the tissue tropism and transmissibility. Following intranasal inoculation of golden Syrian hamsters, nasal turbinates, trachea,

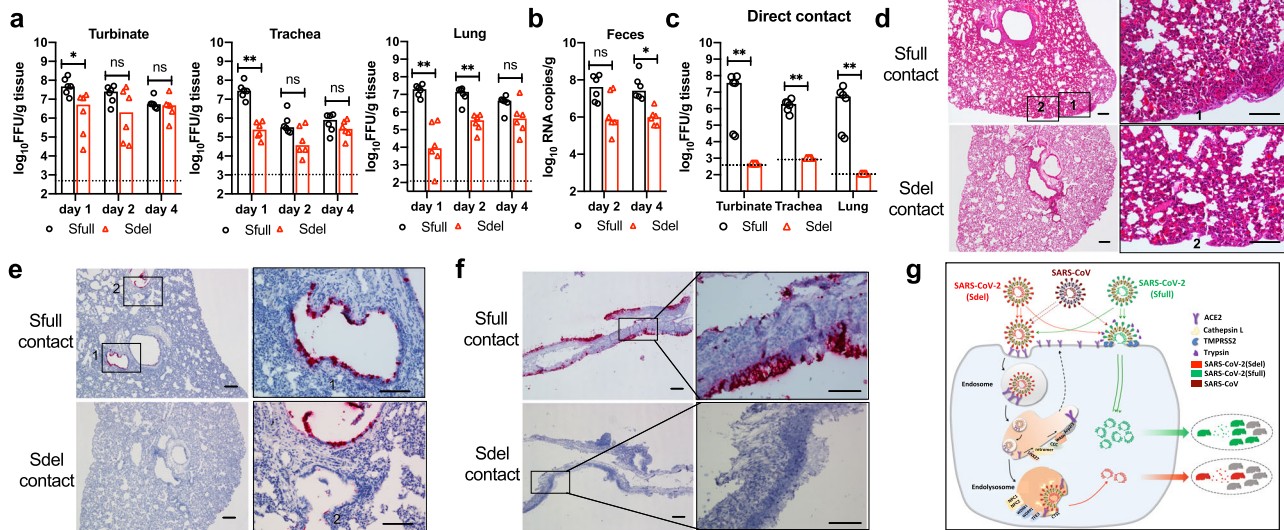

**Fig. 5 The switch of virion entry pathways modulates the infection and transmission in hamsters. a** Viral load in the tissues of nasal turbinate, trachea, and lung. Tissues were harvested at days 1, 2, and 4 post challenge of Sfull or Sdel virus (n = 6 per day). **b** Viral RNA in fecal samples. Fresh fecal samples were collected at days 2 and 4 post infection of Sfull or Sdel strain (n = 6 per day) for qRT-PCR. **c** Transmission of Sfull or Sdel strain in hamsters by direct contact exposure. Naïve hamsters (n = 6) were each co-housed with one inoculated donor at day 1 for 3 days. Hamsters were sacrificed and the indicated tissues were harvested for titration. The dashed lines represent the limit of detection by focus-forming assay. Median viral titers (**a**–**c**): two-tailed Mann–Whitney test; *P < 0.05; **P < 0.01; n.s. not significant. **d** H&E staining of lung sections of contact hamsters. Representative images are shown from n = 6 hamsters. Scale bar, 100 μm. **e**, **f** RNA ISH of lung and nasal turbinate sections of contact hamsters. Representative images are shown from n = 6 hamsters. Scale bar, 100 μm. **g** Model of the role of S1/S2 boundary and host factors in regulating cell entry, pathogenicity, and transmissibility of SARS-CoV-2. SARS-CoV-2 enters cells via two pathways. The virus (Sfull) with intact spike protein preferentially enters cells at the plasma membrane (early entry pathway) in airway epithelial cells (Calu-3) or respiratory tract tissues expressing the proteases (e.g., TMPRSS2) to activate the membrane fusion. The deletion at S1/S2 junction site in spike (Sdel), however, propels the virus to enter cells through the endosomal pathway (late entry pathway), which is less efficient than the fusion pathway at the plasma membrane. Host factors such as CTSL, NCP1/2, WDR81/91, and TFE3 are critical for the endosomal entry. Both entry pathways are initiated with virion binding to receptor ACE2 at the cell surface that is regulated by host factors including retromer, CCC, and WASH complexes, etc. The more efficient early entry pathway in respiratory tract with intact spike protein than the late pathway promotes virus production, pathogenesis, and transmission in a hamster model. The SARS-CoV with spike lacking the insertion of multi-basic amino acids may resemble the Sdel virus and enter cell less efficiently than SARS-CoV-2 resulting in relatively low transmissibility.

lungs, heart, kidney, spleen, duodenum, brain, serum, and feces were collected. Sfull virus replicated robustly and reached peak titer at day 1 post infection, with a mean titer 31-, 126-, and 1259-fold higher than Sdel in the turbinates, trachea, and lungs, respectively (Fig. 5a). While Sdel virus replication was delayed, no significant differences were observed by day 4 in these three tissues. At days 2 and 4, five pieces of fresh feces were collected from each hamster. Although no infectious virus was detected by focus-forming assay, viral RNA levels were higher in fecal samples for Sfull (20- and 40-fold) than Sdel at days 2 and 4, respectively (Fig. 5b). Likely related to this, no infectious virus was detected in the duodenum, and Sfull RNA was 6.3-fold higher than Sdel at day 4 (Supplementary Fig. 7a). In serum, we detected no difference in viremia at day 1, but Sfull RNA was 63- and 32-fold higher than Sdel at days 2 and 4, respectively (Supplementary Fig. 7b). In other extrapulmonary organs, infectious virus was not consistently detected. In general, brain tissue had the highest viral RNA copy number, and all organs showed higher levels of Sfull RNA at day 2 or 4 compared to Sdel except for the liver and kidneys (Supplementary Fig. 7c–g). Body weight of six hamsters challenged with Sfull or Sdel virus were monitored daily, and the weight loss was only observed in hamsters inoculated with Sfull and decreased as much as ~18% at days 5 and 6 (Supplementary Fig. 7h).

To determine the impact of deletion at the S1/S2 junction on transmissibility by direct contact exposure, six hamsters were inoculated intranasally with Sfull or Sdel virus. At 24 h post inoculation, each donor hamster was transferred to a new cage and co-housed with one naïve hamster for 3 days. For donors

(day 4 post inoculation), tissue samples were processed (Fig. 5a, b and Supplementary Fig. 7). For contact hamsters (day 3 post-exposure), nasal turbinate, trachea, and lungs were collected for infectious virus titration and histopathological examination. The average titers in turbinate, trachea, and lungs from Sfull-exposed hamsters reached 6.6, 6.2, and 6.1 logs, respectively (Fig. 5c). Unexpectedly, no infectious virus was detected in these three tissues from Sdel-exposed hamsters (Fig. 5c). In lung sections from hamsters that were exposed to Sfull-infected animals, we observed mononuclear cell infiltrate, protein-rich fluid exudate, hyaline membrane formation, and hemorrhage (Fig. 5d). In contrast, no or minimal histopathological change was observed in the lung sections from hamsters that were exposed to Sdel-infected animals (Fig. 5d). To examine viral spread in the lungs, we performed RNA in situ hybridization (ISH). Viral RNA was clearly detected in bronchiolar epithelial cells in hamsters exposed to Sfull-infected animals (Fig. 5e) whereas it was rarely detected in hamsters exposed to Sdel-infected animals. Similarly, abundant RNA was observed in the nasal turbinate epithelium (Fig. 5f). These results indicated that transmission of Sfull from infected hamsters to co-housed naïve hamsters was efficient whereas the deletion at the S1/S2 boundary in the S protein of Sdel markedly reduced transmission.

## Discussion

Using authentic infectious viruses, our in vitro and in vivo studies establish that the unique S1/S2 boundary of the SARS-CoV-2 S protein can modulate the entry pathways and transmission of the

virus (Fig. 5g). In Calu-3 cells that expresses the TMPRSS2, the Sfull virus with an intact boundary bearing the multi-basic residues, RRAR, preferentially enters cells through the plasma membrane fusion pathway, whereas Sdel with the deletion disrupting these residues propels the cell entry to an endosomal pathway and reduces the infectivity. This is further demonstrated when we mutated two basic residues in the RRAR motif (R682S, R685S), which led to less efficient infection in Calu-3 cells. In Vero cells expressing no or minimal TMPRSS2, Sfull virus enters via endosomal pathway, making the multi-basic residues dispensable, which results in its deletion, presumably due to an adaptive advantage. This deletion effect could be abrogated by adding trypsin or by expressing TMPRSS2, which allows the virus to resume entry via the plasma membrane fusion pathway, as we verified by the acquisition of sensitivity to camostat in Vero-TMPRSS2 cells for Sfull virus. In contrast, the Sdel virus maintains its usage of the E-64d-sensitive endosomal pathway for entry even in Vero-TMPRSS2 cells. It is noteworthy that infection by Sdel virus, but not Sfull, in A549-ACE2 cells is sensitive to the cathepsins B/L inhibitor E-64d, highlighting the importance of S1/S2 boundary sequence in this entry process. Treatment with camostat has no impact on Sfull virus infection in A549-ACE2 cells, as no or minimal TMPRSS2 is expressed, suggesting that other TMPRSS2 homologs or trypsin-like proteases may activate the Sfull virus entry at the plasma membrane. The results of our experiments using the pseudovirus bearing the SARS-CoV spike protein, which lacks the multiple basic residues at the S1/S2 junction, were similar to what we observed for the Sdel virus.

Although the Sfull virus enters cells through both plasma and endosomal fusion pathways in some cells types such as Calu-3 and Vero-TMPRSS2 cells, the specific endosomal entry of Sdel virus in A549 cells has provided a very useful platform for investigating the endosomal entry process. Thus, we conducted the genome-wide CRISPR screen with this platform and uncovered a large number of host factors that regulate the virus entry. Genes for the endosomal entry-specific enzyme CTSL and for regulating endolysomal trafficking and membrane fusion, such as *NPC1/2* and *WDR81/91*, were identified and required for Sdel, but not for Sfull virus infection in A549 cells. In parallel, we discovered a panel of entry factors common to both Sdel and Sfull that regulate the surface expression of the SARS-CoV-2 receptor ACE2. Strikingly, all the genes validated for Sdel virus are also required for the pseudotyped SARS-CoV and MERS-CoV, highlighting the similar entry machinery employed by members of coronavirus family in specific cell type. Understanding the detailed mechanisms of action for these common host factors could help in the development of potential countermeasures to combat COVID-19 or other related coronaviruses. However, it has to be pointed out that, given the two different entry pathways used by SARS-CoV-2 in a cell-type-dependent manner, targeting the endosomal entry pathway only might not be a promising strategy to inhibit SARS-CoV-2 infection. This is exemplified by the in vitro and in vivo results of studies examining the lysosomal acidification inhibitors chloroquine and hydroxychloroquine[49–51]. Likewise, it raises the question of the effectiveness of perturbing cholesterol trafficking with inhibitors such as U18666A targeting the host factor NPC1 in COVID-19 patients as previously proposed[52,53]. Thus, to further dissect the plasma membrane entry pathway and identify the relevant host genes may help to better understand the whole landscape of SARS-CoV-2 entry.

The serine protease TMPRSS2 on the cell surface activates the spike protein-mediated membrane fusion pathway, which is important for virus spread[54,55]. It has been reported that TMPRSS2 is enriched in nasal and bronchial tissues[56–58], implying that the transmission of SARS-CoV-2 by respiratory droplets might be enhanced for virus bearing an intact versus a deleted S1/S2 boundary. In our hamster experiments, the deletion mutant virus Sdel exhibited decreased viral infection and disease compared to Sfull. More importantly, the transmission of Sdel by direct contact exposure for 3 days was almost completely abrogated. The nearly complete abrogation of infection by direct contact highlights the critical role of the multi-basic sequence at the S1/S2 boundary in transmissibility, presumably due to usage of the more efficient plasma membrane fusion entry pathway. It has to be mentioned that transmission of Sdel might be delayed as compared to Sfull. The 3 days of contact exposure could be prolonged to assess the transmissibility. Also, nasal washes and throat swabs could be collected to determine the difference of virus shedding between the Sfull and Sdel.

In summary, we have demonstrated that the sequences at the S1/S2 boundary of SARS-CoV-2 spike protein modulate the entry pathways, infectivity, and transmissibility, and have identified host genes that regulate the viral entry, specifically the endosomal entry pathway, using the spike deletion mutant virus as a tool.

## Methods

**Cells**. Vero E6 (Cell Bank of the Chinese Academy of Sciences, Shanghai, China), HEK 293T (ATCC # CRL-3216), HeLa (ATCC #CCL-2), A549 (ATCC #CCL-185), and Calu-3 (Cell Bank of the Chinese Academy of Sciences, Shanghai, China) all were cultured at 37 °C in Dulbecco's modified Eagle's medium (Hyclone #SH30243.01) supplemented with 10% fetal bovine serum (FBS), 10 mM HEPES, 1 mM sodium pyruvate, 1× non-essential amino acids, and 100 U/ml of penicillin–streptomycin. The A549-ACE2 and HeLa-ACE2 clonal cell lines were generated by transduction of lentivector expressing the human ACE2 gene as described below. Similarly, the bulk Vero-TMPRSS2 cells were generated by transduction of lentivector expressing the human TMPRSS2 and selected with puromycin. The expression of ACE2 or TMPRSS2 was confirmed by flow cytometry or western blotting. All cell lines were tested routinely and free of mycoplasma contamination.

**Viruses**. The SARS-CoV-2 nCoV-SH01 strain (GenBank accession no. MT121215) was isolated from a COVID-19 patient by passaging in Vero E6 cells twice in the presence of trypsin. Collection of the COVID-19 patient samples and the study was approved by the Shanghai Municipal Health and Family Planning Commission. The procedures were carried out in accordance with approved guidelines. This virus stock underwent three rounds of plaque purification in Vero E6 cells in the presence of trypsin and designated as SH01-Sfull (thereafter as Sfull). Sfull stain was then passaged twice and plaque-purified once in the absence of trypsin, resulting the stain Sdel that has 21 nt deletion in the spike gene. Sfull virus was also passaged twice in Vero E6 cells in the presence of trypsin or twice in Vero E6 ectopically expressing the TMPRSS2 without trypsin. The virus titers were titrated in Vero E6 cells in the presence of trypsin by focus-forming assay as described below. The full genome of Sfull and Sdel strains, and the entire spike gene of other passaged viral stocks were Sanger sequenced and analyzed. All the sequencing primers are available upon request. All experiments involving virus infections were performed in the biosafety level 3 (BSL-3) facility of Fudan University following the regulations.

**Genome-wide CRISPR sgRNA screen**. A human Brunello CRISPR knockout pooled library encompassing 76,441 different sgRNAs targeting 19,114 genes[24] was a gift from David Root and John Doench (Addgene #73178), and amplified in Endura competent cells and purified with Plasmid Maxi Kit (Qiagen #12163). The sgRNA plasmid library was packaged in 293FT cells after co-transfection with psPAX2 (Addgene #12260) and pMD2.G (Addgene #12259) at a ratio of 2:2:1 using Fugene®HD (Promega). At 48 h post transfection, supernatants were harvested, clarified by spinning at 3000 r.p.m. for 15 min, and aliquoted for storage at −80 °C.

For the CRISPR sgRNA screen, A549-ACE2-Cas9 cells were generated by transduction of A549-ACE2 cell line with a packaged lentivirus expressing the mCherry derived from the lentiCas9-Blast (Addgene #52962) that the blasticidin resistance gene was replaced by mCherry. The sorted mCherry-positive A549-ACE2-Cas9 cells were transduced with packaged sgRNA lentivirus library at a multiplicity of infection (MOI) of ~0.3 by spinoculation at 1000g and 32 °C for 30 min in 12-well plates. After selection with puromycin for around 7 days, ~1 × 10[8] cells in T175 flasks were inoculated with SARS-CoV-2 Sdel strain (MOI of 3) and then incubated until nearly all cells were killed. The medium was changed and remaining live cells grew to form colonies. The cells were then harvested and re-plated to the flasks. After second round of killing by the virus, the remaining cells were expanded and ~3 × 10[7] of cells were collected for genomic DNA extraction. Genomic DNA from the uninfected cells (5 × 10[7]) was extracted as the control. The sgRNA sequences were amplified[59] and subjected to next-generation sequencing

using an Illumina NovaSeq 6000 platform. The sgRNA sequences targeting specific genes were extracted using the FASTX-Toolkit (http://hannonlab.cshl.edu/fastx_toolkit/) and cutadapt 1.8.1, and further analyzed for sgRNA abundance and gene ranking by a published computational tool (MAGeCK)[60] (see Supplementary Data 1).

**Gene validation**. Top 32 genes from the MAGeCK analysis were selected for validation. Two independent sgRNAs per gene were chosen from the Brunello CRISPR knockout library and cloned into the plasmid lentiCRISPR v2 (Addgene #52961) and packaged with plasmids psPAX2 and pMD2.G. A549-ACE2, HeLa-ACE2, or Calu-3 cells were transduced with lentiviruses expressing individual sgRNA and selected with puromycin for 7 days. The gene-edited mixed population of cells was used for all the experiments in this study. The sgRNA sequences used for gene editing are listed in Supplementary Data 2.

For virus infection, gene-edited A549-ACE2 or HeLa-ACE2 cells were inoculated with Sfull (MOI 2) and Sdel (MOI 2). Vero, Vero-TMPRSS2, and Calu-3 cells were inoculated with Sfull (MOI 1) and Sdel (MOI 1). At 24 h post infection, cells were fixed with 4% paraformaldehyde (PFA) diluted in phosphate-buffered saline (PBS) for 30 min at room temperature, and permeabilized with 0.2% Triton x-100 in PBS for 1 h at room temperature. Cells then were subjected for immunofluorescence staining and imaging as described below. Validation also was performed by an infectious virus yield assay.

**Virus yield assay**. Calu-3 cells were seeded one day prior to infection. Cells were inoculated with same MOI of Sfull or Sdel (MOI 0.1) for 1 h. After three times of washing, cells were maintained in 2% FBS culture media, and supernatants were collected at specific time points for titration on Vero cells by focus-forming assay.

**Pseudotyped virus experiment**. Pseudoviruses were packaged in HEK 293T cells by co-transfecting the retrovector pMIG (kindly provided by Jianhua Li, Fudan Univeristy) for which the gene of target was replaced by the nanoluciferase gene, plasmid expressing the MLV Gag-Pol, and pcDNA3.1 expressing different spike genes or VSV-G (pMD2.G (Addgene #12259)) using Fugene®HD transfection reagent (Promega). At 48 h post transfection, the supernatant was harvested, clarified by spinning at 3500 r.p.m. for 15 min, aliquoted, and stored at −80 °C for use. The virus entry was assessed by transduction of pseudoviruses in gene-edited cells in 96-well plates. After 48 or 72 h, the luciferase activity was determined using Nano-Glo® Luciferase Assay kit (Promega #N1110) according to the manufacturer's instructions. The same volume of assay reagent was added to each well and shake for 2 min. After incubation at room temperature for 10 min, luminescence was recorded by using a FlexStation 3 (Molecular Devices) with an integration time of 1 s per well.

**Plasmid construction**. To construct the lentivector expressing the human ACE2 gene, the human ACE2 gene (Miaolingbio #P5271) was PCR-amplified and cloned into the pLV-EF1a-IRES-blast (Addgene #85133). The human TMPRSS2 (Sino Biological #HG13070-CM) and DPP4 (kindly provided by Yaowei Huang, Zhejiang University) genes were cloned by the similar strategy. To construct the vectors for pseudovirus packaging, the full-length spike gene was PCR-amplified from Sfull or Sdel strain and cloned into the pcDNA3.1 vector. The Sfull spike gene with two mutations (R682S, R685S)[16] in the furin cleavage site was generated by PCR. The full-length SARS-CoV or MERS-CoV spike gene was cloned similarly. The primers used for plasmid construction are listed in Supplementary Data 3.

**Virus binding and internalization assays**. A549-ACE2 gene-edited cells were seeded in 24-well plate one day prior to the assays. Plates were pre-incubated on ice for 10 min, then washed twice with ice-cold PBS. Ice-cold Sfull virus (MOI of 5) in a 0.5-ml medium was incubated with cells on ice for 45 min. After five cycles of washing, cells were lysed in TRIzol reagent (Thermo Fisher #15596018) for RNA extraction. For internalization assay, after five cycles of washing, cells were incubated into medium supplemented with 2% FBS and then incubated at 37 °C for 45 min. Cells were chilled on ice, washed with ice-cold PBS, and then treated with 400 µg/ml protease K on ice for 45 min. After three additional washes, cells were lysed in TRIzol reagent for RNA extraction. Reverse transcriptase PCR (RT-qPCR) was conducted to quantify the viral specific nucleocapsid RNA and an internal control GAPDH.

**Cell-based S1-Fc and anti-ACE2 antibody-binding assay**. A549-ACE2 gene-edited cells were seeded in a 96-well plate one day prior to the experiment. Cells were collected with TrypLE (Thermo #12605010) and washed twice with ice-cold PBS. Live cells were incubated with the recombinant protein, S1 domain of SARS-CoV-2 spike C-terminally fused with Fc (Sino Biological #40591-V02H, 1 µg/ml), or the anti-ACE2 antibody (Sino Biological #10108-RP01, 1 µg/ml) at 4 °C for 30 min. After washing, cells were stained with goat anti-human IgG (H + L) conjugated with Alexa Fluor 647 (Thermo #A21445, 2 µg/ml) for 30 min at 4 °C. After two additional washes, cells were subjected to flow cytometry analysis (Thermo, Attune™ NxT) and data processing (FlowJo v10.0.7). The gating strategy was indicated in Supplementary Fig. 8.

**Western blotting**. Cells in plates washed twice with ice-cold PBS and lysed in RIPA buffer (Cell Signaling #9806S) with a cocktail of protease inhibitors (Sigma-Aldrich # S8830). Samples were prepared in reducing buffer (50 mM Tris, pH 6.8, 10% glycerol, 2% SDS, 0.02% [wt/vol] bromophenol blue, 100 mM DTT). After heating (95 °C, 10 min), samples were electrophoresed in 10% SDS polyacrylamide gels, and proteins were transferred to PVDF membranes. Membranes were blocked with 5% non-fat dry powdered milk in TBST (100 mM NaCl, 10 mM Tris, pH 7.6, 0.1% Tween 20) for 1 h at room temperature, and probed with the primary antibodies at 4 °C overnight. After washing with TBST, blots were incubated with horseradish peroxidase (HRP)-conjugated secondary antibodies for 1 h at room temperature, washed again with TBST, and developed using SuperSignal West Pico or Femto chemiluminescent substrate according to the manufacturer's instructions (Thermo Fisher). The antibodies used are as follows: rabbit anti-COMMD3 (proteintech #26240-1-AP, 1:800), rabbit anti-VPS35 (proteintech #10236-1-AP,1:500), rabbit anti-CCDC22 (proteintech #16636-1-AP, 1:1000), rabbit anti-NPC1 (proteintech #13926-1-AP, 1:1000), rabbit anti-NPC2 (proteintech #19888-1-AP, 1:800), rabbit anti-CCDC53 (proteintech #24445-1-AP, 1:500), rabbit anti-COMMD1 (proteintech #11938-1-AP, 1:2000), mouse anti-SNX27 (Abcam #ab77799, 1:1000), rabbit anti-SNX17 (proteintech, #10275-1-AP, 1:2000), rabbit anti-LDLR (proteintech, #10785-1-AP, 1:1000), rabbit anti-LRP1 (Abcam #ab92544, 1:5000), rabbit anti-SARS-Cov-2 spike S2 (Sino Biological #40590-T62, 1:1000), rabbit anti-β-actin (proteintech #20536-1-AP, 1:2000). The HRP-conjugated secondary antibodies include goat anti-mouse (Sigma #A4416, 1:5000), goat anti-rabbit (Thermo Fisher #31460, 1:5000), goat anti-human (Sigma #A6029, 1:5000).

For quantification studies, after probing with primary antibodies, membranes were incubated with goat anti-rabbit IRDye 800CW secondary antibody (LI-COR #926-32211, 1:10,000), goat anti-rabbit IRDye 680RD secondary antibody (LI-COR #926-68071, 1:10,000), or goat anti-mouse IRDye 800CW secondary antibody (LI-COR #926-32210, 1:10,000), then developed and analyzed with the Odyssey CLx Imaging System and Image Studio 4.0 software.

**Biotinylation of plasma membrane proteins**. Gene-edited A549-ACE or Calu-3 cells seeded in six-well plate 24 h prior to experiment were chilled on ice for 10 min, and labeled with 2.5 mg/ml biotin (Thermo Fisher #21331) in PBS for 30 min on ice. Cells were quenched with 100 mM glycine in PBS three times, 10 min each. After washing with PBS, cells were lysed in RIPA buffer (Cell Signaling #9806S) with a cocktail of protease inhibitors (Sigma-Aldrich # S8830), and immunoprecipitated with Streptavidin agarose beads overnight at 4 °C. Beads were then washed three times with RIPA buffer, and eluted into 5× loading buffer (Beyotime #P0015L) at 95 °C for 10 min. After spinning at maximum speed for 10 min, the supernatants were harvested for western blotting using rabbit anti-ACE2 (Abcam #ab15348, 1:1000) as described above, and analyzed with the Odyssey CLx Imaging System and Image Studio 4.0 software. The un-immnoprecipitated lysates were used as a loading control.

**Immunofluorescence assay**. Virus-infected cells were washed twice with PBS, fixed with 4% PFA in PBS for 30 min, permeablized with 0.2% Triton X-100 for 1 h. Cells were then incubated with house-made mouse anti-SARS-CoV-2 nucleocapsid protein serum (1:1000) at 4 °C overnight. After three washes, cells were incubated with the secondary goat anti-mouse antibody conjugated with Alexa Fluor 555 (Thermo #A-21424, 2 µg/ml) for 2 h at room temperature, followed by staining with 4′,6-diamidino-2-phenylindole. Images were collected using an Operetta High Content Imaging System (PerkinElmer), and processed using the PerkinElmer Harmony high-content analysis software v4.9 and ImageJ v2.0.0 (http://rsb.info.nih.gov/ij/).

**Cell viability assay**. A CellTiter-Glo® Luminescent Cell Viability Assay (Promega # G7570) was performed according to the manufacturer's instructions. The same number of gene-edited cells was seeded into opaque-walled 96-well plates. Forty-eight hours later, CellTiter-Glo® reagent was added to each well and allowed to shake for 2 min. After incubation at room temperature for 10 min, luminescence was recorded by using a FlexStation 3 (Molecular Devices) with an integration time of 0.5 s per well.

**Animal experiments**. Six- to ten-week-old male hamsters were used in the study in the BSL-3 laboratory of Fudan University. The experiment protocol has been approved by the Animal Ethics Committee of School of Basic Medical Sciences at Fudan University. The hamsters were inoculated intranasally with $5 \times 10^4$ focus-forming unit of Sfull or Sdel virus. To evaluate the viral transmission by direct contact, at day 1 post infection, each hamster infected with Sfull or Sdel was transferred to a new cage and co-housed with one age-matched naïve hamster for 3 days. At 24, 48, and 96 h post virus challenge, or 72 h post contact, animals were euthanized and the sera were collected. After perfusion extensively with PBS, indicated tissues were harvested for virus titration by focus-forming assay in the presence of trypsin or for histopathological examination. To collect fecal samples, at 48 and 96 h post challenge, each hamster was put into an individual clean container and fresh fecal samples (5 pieces) were collected and frozen down for virus titration by focus-forming assay or RT-qPCR analysis. To monitor the

body weight change, six hamsters were measured daily for 14 days. Tissues were homogenized in DMEM and virus was titrated by focus-forming assay[61] using the rabbit polyclonal antibody against SARS-CoV nucleocapsid protein (Rockland, 200-401-A50, 0.5 µg/ml) or by RT-qPCR after RNA extraction as described below.

**Histology and RNA ISH**. Virus-infected hamsters were euthanized and perfused extensively with PBS. Nasal turbinate and lung tissues were harvested and fixed in 4% PFA for 48 h. Tissues were embedded in paraffin for sectioning and stained with hematoxylin and eosin (H&E) to assess tissue morphology. To determine sites of virus infection, RNA ISH was performed using the RNAscope 2.5 HD Assay (Red Kit) according to the manufacturer's instructions (Advanced Cell Diagnostics). In brief, sections were deparaffinized, treated with $H_2O_2$ and Protease Plus prior to probe hybridization. A probe specifically targeting the SARS-CoV-2 spike RNA (Advanced Cell Diagnostics, #848561) was used for ISH experiments. Tissues were counterstained with Gill's hematoxylin. Tissue sections were visualized using a Nikon Eclipse microscope.

**qRT-PCR**. RNA from serum, tissues, or cells was extracted with the TRIzol reagent (Thermo Fisher #15596018). Viral or host RNA levels were determined using the TaqPath™ 1-Step RT-qPCR Master Mix (Thermo Fisher # A15299) on CFX Connect Real-Time System (Bio-Rad) instrument. A standard curve was produced using serial 10-fold dilutions of in vitro-transcribed RNA of N gene driven by the SP6 promoter (Thermo Fisher #AM1340). Viral burden was expressed on a $\log_{10}$ scale as viral RNA copies per g of tissue or ml of serum. Primers and probes used are as follows: nCoV-N-Fwd: 5′-GACCCCAAAATCAGCGAAAT-3′; nCoV-N-Rev: 5′-TCTGGTTACTGCCAGTTGAATCTG-3′; nCoV-N-Probe: 5′-FAM-ACC CCGCATTACGTTTGGTGGACC-BHQ1-3′; hGAPDH-Fwd: 5′- TGCCTTCT TGCCTCTTGTCT-3′; hGAPDH-Rev: 5′- GGCTCACCATGTAGCACTCA-3′; and GAPDH-Probe: 5′-FAM-TTTGGTCGTATTGGGCGCCTGG-BHQ1-3′.

**Virus load determination by focus-forming assay**. The experiment was performed similarly as described previously[62]. Briefly, Vero E6 monolayer in 96-well plates was inoculated with serially diluted virus for 2 h and then overlaid with methylcellulose for 48 h. Cells were fixed with 4% PFA in PBS for 1 h and permeablized with 0.2% Triton X-100 for 1 h. Cells were stained with rabbit polyclonal antibody against SARS-CoV nucleocapsid protein (Rockland, 200-401-A50, 0.5 µg/ml) overnight at 4 °C, incubated with the secondary goat anti-rabbit HRP-conjugated antibody for 2 h at room temperature. The focus-forming unit was developed using TrueBlue substrate (Sera Care #5510-0030).

**Statistical analysis**. Statistical significance was assigned when $P$ values were <0.05 using Prism Version 8 (GraphPad). Data analysis was determined by a Mann–Whitney, or ANOVA, or unpaired $t$-test depending on data distribution and the number of comparison groups.

**Reporting summary**. Further information on research design is available in the Nature Research Reporting Summary linked to this article.

## Data availability
The authors declare that all relevant data supporting the findings of this study are available within the paper and its Supplementary Information. The Supplementary Data provide information for the CRISPR-Cas9 screen, statistical analysis. Source data are provided with this paper. The full-length genome sequence of SARS-CoV-2 nCoV-SH01 strain is deposited in GenBank (accession no. MT121215). Source Data are provided with this paper.

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

## Acknowledgements

Grants from the National Key Research and Development Program of China (2020YFA0707701 to R.Z.), National Natural Science Foundation of China (32041005 to R.Z.), Project of Novel Coronavirus Research of Fudan University (to Y.X.), Development Programs for COVID-19 of Shanghai Science and Technology Commission (20431900401), Natural Science Foundation of Shanghai (19ZR1470400 to X.H.), and Natural Science Foundation of Guangdong Province (20202001261150000005 to W.X.) supported this work. We thank Prof. Michael S. Diamond (Washington University) for discussions and editorial comments on the manuscript. We also thank Prof. Bin Zhou (Nanjing Agricultural University) and Yaowei Huang (Zhejiang University) for providing key reagents. We wish to acknowledge colleagues at the Biosafety Level 3 Laboratory of Fudan University for help with experiment design and technical assistance.

## Author contributions

Yunkai Zhu, F.F., G.H., Y.W., Y.Y., Yuanfei Zhu, W.X. (Fudan University), R.Z. performed the experiments. Yunkai Zhu, F.F., G.H., Y.W., and R.Z. designed the experiments. X.C., Z.S., W.H., R.Y., D.Q., Q.D., X.H., H.C., W.X. (Southern Medical University), Y.X., Q.C., and Z.Y. provided administrative, supervision, technical, or material support. Yunkai Zhu, F.F., G.H., Y.W., Y.Y., and R.Z. performed data analysis. R.Z. wrote the initial draft of the manuscript, with the other authors contributing to editing into the final form.

## Competing interests

The authors declare no competing interests.
