## [Peer Review File · Nature Communications]

Reviewers' Comments:

Reviewer #1:

Remarks to the Author:

The manuscript entitled "A genome-wide CRISPR screen identifies host factors that regulate the SARS-CoV-2 entry" by Zhu and colleagues is a thorough investigation of the functional role played by the multi-basic motif found at the S1/S2 boundary of the SARS-CoV-2 spike (S) protein. Like other groups, the authors describe the emergence of an S1/S2 deletion variant (named Sdel) after a few passages of a patient-derived strain of SARS-CoV-2 (parental strain, named Sfull) in Vero E6 cells. Using authentic virus and pseudotyped virus infections in various cell lines, the authors show a marked difference in infectivity between Sfull and Sdel viral strains. The S1/S2 deletion variant strain displayed enhanced infectivity over the Sfull strain in most cell lines tested except for Calu-3 cells. Zhu and co-workers assessed dependency both strains for cell surface expressed TMPRSS2 protease or endosomal cathepsins using pharmacological inhibitors in different cell types. They show that the Sfull strain is preferentially endocytosed in cells that do not express TMPRSS proteases such as Vero cells. The deletion of the S1/S2 basic motif appears to switch virus entry to become more dependent on the endocytic route compared to entry at the plasma membrane in A549 and Calu-3 cells. The authors then used the Sdel strain to infect A549 cells and performed a genome-wide CRISPR/Cas9 screen to determine host factors involved in the endosomal route of entry. They validated their approach by demonstrating that the ACE2 receptor was the top hit in their screen and show also the importance of cathepsin L, confirming earlier findings with pharmacological inhibition of the protease. The authors further characterized the top 32 hits from their screen and show that they belonged to 4 broad categories of endosomal cargo sorting complexes: retromer complex, COMMD/CCDC22/CCDC93 (CCC) complex, Wiskott-Aldrich syndrome protein and SCAR homologue (WASH) complex, and the actin-related protein 2/3 (Arp2/3 complex). Interestingly the screen also revealed that Sdel infection depended on NPC1 and NPC2 cholesterol transporters. The top hits were validated using pseudovirus infection assays with pseudovirions bearing SARS-CoV-2 S from Sdel strain, VSV G, SARS-CoV S and MERS-CoV S. Authors confirmed the importance of these host factors for SARS-CoV-2 Sdel pseudovirion entry. VSV G pseudovirion appeared insensitive to the editing of a majority of genes identified in the screen. Remarkably, SARS-CoV S pseudovirions exhibited similar dependence of host factors than the Sdel strain pseudovirions. In contrast infection with authentic Sfull strain virus displayed a different pattern of dependency of the host factors identified in the screen, with a decrease in the dependency for protein complexes involved in endosomal retrieval and recycling. The authors also showed that the host factors involved in retrieval and recycling of endosomes also play a role in regulating the expression of ACE2 at the cell surface, which directly affects SARS-CoV-2 entry and infection. The authors also analyzed other components of the endosomal cargo sorting complexes that were not identified in the screen and confirm the importance of these complexes for SARS-CoV-2 entry. Finally authors performed a series of in vivo infection experiments using Sdel and Sfull viruses and golden Syrian hamsters as animal model. They demonstrate convincingly that at early time points Sfull strain replicated more robustly than Sdel, particularly in the lungs. Weight loss was apparent only in Sfull strain-inoculated animals. The authors performed transmissibility experiments by co-housing Sdel or Sfull infected animals with naïve ones and show that while the Sfull strain could readily transmit, the Sdel strain displayed markedly reduced transmission towards uninfected hamsters.

This is a very comprehensive and well-conceived study that sheds light on the functional role played by the S1/S2 multi-basic motif of SARS-CoV-2 spike protein. Importantly, the genome-wide CRISPR/Cas9 screen provides important information on the role played by endosomal sorting complexes in the entry of SARS-CoV-2. In addition the in vivo experiments demonstrate convincingly that the S1/S2 multi-basic motif is critical for virus spread and transmission. The authors provide a wealth of solid and relevant data that are convincingly presented and are of interest to a wide audience. Below are a few minor points and suggestions that would help clarify certain aspects of the manuscript.

Minor comments

1. In figure 3 it is not very clear why authors do not show data for Sfull pseudovirions and instead present data only for Sfull authentic virus infection. Showing Sfull pseudovirion data would allow to directly compare results with those of Sdel pseudovirions. If the data for Sfull pseudovirions is not available for technical reasons, it would be useful for authors to briefly explain why they chose to show only Sfull authentic virus infection.

2. Below are a few typos and suggestions to improve the manuscript:

- Page 3 line 63: consider changing "Coronavirus enters" with "Coronaviruses enter"
- Page 3 lines 63-76: in the last paragraph of the introduction the authors do not introduce the fact that they have performed a genome-wide CRISPR/Cas9 screen. This is a very important aspect of their work and should be stated here.
- Page 8 line 197: consider revising the order of words from "these genes edited" to "these edited genes"
- Page 14 line 336: "Similaly" should be changed to "Similarly"
- Page 14 lines 348-350: "This is further demonstrated when we mutated two basic residues in the RRAR motif (R682S, R685S), which led to less efficient infection of Sdel." It is unclear how can residues within the RRAR motif can be mutated if they have already been deleted in Sdel? Please clarify the wording.

Reviewer #2:

Remarks to the Author:

Zhu and colleagues investigate how a full or deleted multibasic cleavage site in the SARS CoV-2 S protein affects the use of entry pathways and they study the enzymes, endolysosomal trafficking regulators and other host factors involved in these pathways. Subsequently, the direct contact transmissibility of Sdel and Sfull was studied in hamsters.

The manuscript is well written and the experiments are scientifically solid. However, some clarification is needed with respect to the hamsters experiments. It is clear to me now (after some puzzling) that the donors in the transmission experiment are the same animals as the day 4 animals, but it is unclear where the day 14 animals come from and what the group size is.

Line 326: the peak titers, are those the titers of the animal with the highest titer at that timepoint? Usually peak titers refer to titers on the day that the titers were the highest per group. It is not informative the way the titers are presented now, because we may just look at outliers. The average titer per group per day is enough here.

Unfortunately nasal washes (and throat swabs) were not collected from the hamsters, which I strongly recommend for future experiments (light inhalation anesthesia, 500 ul of PBS added dropwise to the nose (hamster positioned head down), collection of 'sneezes'/exhaled PBS in a 10 cm dish). The authors raise the point in the discussion that TMPRSS2 is enriched in nasal and bronchial tissues, and is thus likely to affect virus shedding of Sdel versus Sfull, however no virus shedding kinetics are presented in the manuscript. This would have been an added value since titers in nasal turbinates do not necessarily represent excreted virus titers.

In addition, removal of recipient animals at day 3 is a bit early. Maybe transmission of Sdel is delayed as compared to Sfull, but you don't know now. Please comment on this in the discussion. Next time follow donor and recipients longer and collect swabs daily or every other day to know when transmission happened. This allows you to study delay in transmission as well as seroconversion on a later time point.

Minor points:

- Ref 3 is misplaced. This study is not about transmission.
- Ref 17 is not formatted correctly.

Reviewer #3:

Remarks to the Author:

In this manuscript, the authors generated a pair of SARS-CoV-2 viruses that differ in the multibasic cleavage site in spike: Sfull has an intact cleavage site, whereas Sdel carries a deletion of the cleavage site. The authors characterize the entry of these two viruses in a set of cell lines and then perform a genome-wide CRISPR/cas9 screen with the Sdel virus in A549-ACE2 cells. Several of the identified proviral hits are analyzed for their function in viral entry. Lastly, the authors compare the pathogenicity and transmissibility of Sfull and Sdel in a hamster model.

The topic is of current interest and most of the experiments are well controlled, sufficiently described and carefully discussed. The findings add important novel pieces to our understanding of SARS-CoV-2 entry and are thus of interest to a broad audience. However, a number of points should be addressed to improve the manuscript:

- The abstract does not convey the key findings in a clear manner. Too many abbreviations are used and there is no clear structure or flow. The abstract should thus be revised.
- Fig. 1: The authors should visualize the different entry routes (early versus late) in the different settings, e.g. by co-staining of viral proteins with endosomal markers or by performing fusion assays based on lipophilic dyes. The idea that SARS-CoV-2 can use two different entry routes is not novel but providing clear evidence for this and comparing the pathways in the different cell lines could be a major selling point for this manuscript. The inhibitor data already support the concept but visualizing the two entry routes would add another level of confirmation.
- L.89-91: How many independent passaging attempts were made in Vero - trypsin, Veros + trypsin and in Vero-TMPRSS2? Are the results shown in fig. S1 from just one experiment? Then the authors should be careful with interpretation. If they want to conclude that the presence of trypsin or TMPRSS2 impacts the selection of a deletion variant multiple passaging attempts need to be performed in parallel.
- Fig. 4A: In the results section it is not described how virus binding and internalization were measured.
- Fig. 5: If the virus replicates 1-2 logs less it is not surprising that transmission does not work. So to me it is not clear what I am learning from the hamster data and what they add to the manuscript.
- Ref. 17 Something went wrong with reference 17. No paper is listed.

Response to Reviewers:

Reviewer #1:

The manuscript entitled “A genome-wide CRISPR screen identifies host factors that regulate the SARS-CoV-2 entry” by Zhu and colleagues is a thorough investigation of the functional role played by the multi-basic motif found at the S1/S2 boundary of the SARS-CoV-2 spike (S) protein. Like other groups, the authors describe the emergence of an S1/S2 deletion variant (named Sdel) after a few passages of a patient-derived strain of SARS-CoV-2 (parental strain, named Sfull) in Vero E6 cells. Using authentic virus and pseudotyped virus infections in various cell lines, the authors show a marked difference in infectivity between Sfull and Sdel viral strains. The S1/S2 deletion variant strain displayed enhanced infectivity over the Sfull strain in most cell lines tested except for Calu-3 cells. Zhu and co-workers assessed dependency both strains for cell surface expressed TMPRSS2 protease or endosomal cathepsins using pharmacological inhibitors in different cell types. They show that the Sfull strain is preferentially endocytosed in cells that do not express TMPRSS proteases such as Vero cells. The deletion of the S1/S2 basic motif appears to switch virus entry to become more dependent on the endocytic route compared to entry at the plasma membrane in A549 and Calu-3 cells. The authors then used the Sdel strain to infect A549 cells and performed a genome-wide CRISPR/Cas9 screen to determine host factors involved in the endosomal route of entry. They validated their approach by demonstrating that the ACE2 receptor was the top hit in their screen and show also the importance of cathepsin L, confirming earlier findings with pharmacological inhibition of the protease. The authors further characterized the top 32 hits from their screen and show that they belonged to 4 broad categories of endosomal cargo sorting complexes: retromer complex, COMMD/CCDC22/CCDC93 (CCC) complex, Wiskott-Aldrich syndrome protein and SCAR homologue (WASH) complex, and the actin-related protein 2/3 (Arp2/3 complex). Interestingly the screen also revealed that Sdel infection depended on NPC1 and NPC2 cholesterol transporters. The top hits were validated using pseudovirus infection assays with pseudovirions bearing SARS-CoV-2 S from Sdel strain, VSV G, SARS-CoV S and MERS-CoV S. Authors confirmed the importance of these host factors for SARS-CoV-2 Sdel pseudovirion entry. VSV G pseudovirion appeared insensitive to the editing of a majority of genes identified in the screen. Remarkably, SARS-CoV S pseudovirions exhibited similar dependence of host factors than the Sdel strain pseudovirions. In contrast infection with authentic Sfull strain virus displayed a different pattern of dependency of the host factors identified in the screen, with a decrease in the dependency for protein complexes involved in endosomal retrieval and recycling. The authors also showed that the host factors involved in retrieval and recycling of endosomes also play a role in regulating the expression of ACE2 at the cell surface, which directly affects SARS-CoV-2 entry and infection. The authors also analyzed other components of the endosomal cargo sorting complexes that were not identified in the screen and confirm the importance of these complexes for SARS-CoV-2 entry. Finally authors performed a series of in vivo infection experiments using Sdel and Sfull viruses and golden Syrian hamsters as animal model. They demonstrate convincingly that at early time points Sfull strain replicated more robustly than Sdel, particularly in the lungs. Weight loss was apparent only in Sfull strain-inoculated animals. The authors performed transmissibility experiments by co-housing Sdel or Sfull infected animals with naïve ones and show that while the Sfull strain could readily transmit, the Sdel strain displayed markedly reduced transmission towards uninfected hamsters.

This is a very comprehensive and well-conceived study that sheds light on the functional role played by the S1/S2 multi-basic motif of SARS-CoV-2 spike protein. Importantly, the genome-wide CRISPR/Cas9 screen provides important information on the role played by endosomal sorting complexes in the entry of SARS-CoV-2. In addition the in vivo experiments demonstrate convincingly that the S1/S2 multi-basic motif is critical for virus spread and transmission. The authors provide a wealth of solid and relevant data that are convincingly presented and are of interest to a wide audience. Below are a few minor points and suggestions that would help clarify certain aspects of the manuscript.

We greatly appreciate the favorable summary of our work.

Minor comments

1. In figure 3 it is not very clear why authors do not show data for Sfull pseudovirions and instead present data only for Sfull authentic virus infection. Showing Sfull pseudovirion data would allow to directly compare results with those of Sdel pseudovirions. If the data for Sfull pseudovirions is not available for technical reasons, it would be useful for authors to briefly explain why they chose to show only Sfull authentic virus infection.

Thanks for the Reviewer's comment. Initially, the purpose of Fig. 3a and 3b was to determine whether the host genes identified were required for virus entry using the pseudovirus (retrovector-based) bearing the spike protein of SARS-CoV-2 or an unrelated glycoprotein of VSV. Here we used the spike protein of Sdel strain to package the pseudovirus as the live Sdel virus was already used to validate the genes in Fig. 2c. We then attempted to package the pseudovirus using the spike protein of Sful virus. However, the titer (luminescence signal) is pretty low, just around 1,000-3,000 RLU, resulting in variations across different experiments. It was difficult to draw a clear conclusion, especially when there were over 30 gene-edited cell lines to test each time. Since we used the authentic Sfull virus to verify the genes in Fig. 3e in comparison to live Sdel virus in Fig. 2c, we gave up the experiment with the pseudovirus bearing the Sfull spike protein. It is interesting to notice that the spike protein with deletion or mutation at the S1/S2 boundary disrupting the multibasic site can package higher pseudovirus titer, as reported by other groups. Such phenomenon might be cell-type specific.

2. Below are a few typos and suggestions to improve the manuscript:

- Page 3 line 63: consider changing "Coronavirus enters" with "Coronaviruses enter"

We have corrected this error.

- Page 3 lines 63-76: in the last paragraph of the introduction the authors do not introduce the fact that they have performed a genome-wide CRISPR/Cas9 screen. This is a very important aspect of their work and should be stated here.

We appreciate the suggestion, and have added this (page 3 line107-109).

- Page 8 line 197: consider revising the order of words from "these genes edited" to "these edited genes"

We have corrected this error.

- Page 14 line 336: "Similaly" should be changed to "Similarly"

We have corrected this error.

- Page 14 lines 348-350: "This is further demonstrated when we mutated two basic residues in the RRAR motif (R682S, R685S), which led to less efficient infection of Sdel." It is unclear how can residues within the RRAR motif can be mutated if they have already been deleted in Sdel? Please clarify the wording.

We apologize for the confusion, and have corrected this error.

Reviewer #2:

Zhu and colleagues investigate how a full or deleted multibasic cleavage site in the SARS CoV-2 S protein affects the use of entry pathways and they study the enzymes, endolysosomal trafficking regulators and other host factors involved in these pathways. Subsequently, the direct contact transmissibility of Sdel and Sfull was studied in hamsters.

The manuscript is well written and the experiments are scientifically solid. However, some clarification is needed with respect to the hamsters experiments. It is clear to me now (after some puzzling) that the donors in the transmission experiment are the same animals as the day 4 animals, but it is unclear where the day 14 animals come from and what the group size is.

We appreciate the comment and apologize for the confusion. Six hamsters per group challenged with Sfull or Sdel virus were maintained for 14 days, and the body weight was monitored daily. We have clarified this in the paragraph of "Animal experiments" in "Methods" section, and also in the main text (page 13 line 440-441)

Line 326: the peak titers, are those the titers of the animal with the highest titer at that timepoint? Usually peak titers refer to titers on the day that the titers were the highest per group. It is not informative the way the titers are presented now, because we may just look at outliers. The average titer per group per day is enough here.

We apologize for this. We have corrected tissue titers as "average titers" (page 13 line 450-451).

Unfortunately nasal washes (and throat swabs) were not collected from the hamsters, which I strongly recommend for future experiments (light inhalation anesthesia, 500 ul of PBS added dropwise to the nose (hamster positioned head down), collection of 'sneezes'/exhaled PBS in a 10 cm dish). The authors raise the point in the discussion that TMPRSS2 is enriched in nasal and bronchial tissues, and is thus likely to affect virus shedding of Sdel versus Sfull, however no virus shedding kinetics are presented in the manuscript. This would have been an added value since titers in nasal turbinates do not necessarily represent excreted virus titers.

We appreciate the suggestion and agree with the Reviewer that nasal washes and throat swabs are worth collecting, because the virus spread is mainly through the respiratory droplets. Our transmission experiment was set up by direct contact exposure. Even under such condition, we detected no live virus in the contacted hamsters, strongly suggesting the important role of S1/S2 boundary site in transmission. In addition, fecal-oral route could not be excluded in our experiment as hamsters eat their own feces. We agree that collecting the nasal washes (and throat swabs) could be an added value and

we will do this for future experiments. We also greatly appreciate the detailed protocol provided by the Reviewer.

In addition, removal of recipient animals at day 3 is a bit early. Maybe transmission of Sdel is delayed as compared to Sfull, but you don't know now. Please comment on this in the discussion. Next time follow donor and recipients longer and collect swabs daily or every other day to know when transmission happened. This allows you to study delay in transmission as well as seroconversion on a later time point.

We totally agree with the Reviewer's comment that 3 days of exposure is a bit short and the transmission might be delayed. Also, higher doses of virus could be used to challenge the hamsters to test the transmission. We have added this comment in the discussion (page 16 line 547-550).

Minor points:

-Ref 3 is misplaced. This study is not about transmission.

We have corrected this error.

-Ref 17 is not formatted correctly.

We have corrected this error.

Reviewer #3:

In this manuscript, the authors generated a pair of SARS-CoV-2 viruses that differ in the multibasic cleavage site in spike: Sfull has an intact cleavage site, whereas Sdel carries a deletion of the cleavage site. The authors characterize the entry of these two viruses in a set of cell lines and then perform a genome-wide CRISPR/cas9 screen with the Sdel virus in A549-ACE2 cells. Several of the identified proviral hits are analyzed for their function in viral entry. Lastly, the authors compare the pathogenicity and transmissibility of Sfull and Sdel in a hamster model.

The topic is of current interest and most of the experiments are well controlled, sufficiently described and carefully discussed. The findings add important novel pieces to our understanding of SARS-CoV-2 entry and are thus of interest to a broad audience.

We appreciate this favorable and supportive comment.

However, a number of points should be addressed to improve the manuscript:

- The abstract does not convey the key findings in a clear manner. Too many abbreviations are used and there is no clear structure or flow. The abstract should thus be revised.

We have modified the abstract.

- Fig. 1: The authors should visualize the different entry routes (early versus late) in the different settings, e.g. by co-staining of viral proteins with endosomal markers or by performing fusion assays based on lipophilic dyes. The idea that SARS-CoV-2 can use two different entry routes is not novel but providing clear evidence for this and comparing the pathways in the different cell lines could be a major selling point for this manuscript. The inhibitor data already support the concept but visualizing the two entry routes would add another level of confirmation.

We agree with the Reviewer that visualizing the entry routes in different cell types could further confirm the idea of dual entry pathway. We performed the internalization assays in A549-ACE2 or Calu-3 cells with both Sfull and Sdel viruses. The cells were fixed, permeabilized, and stained with viral nucleocapsid protein or endolysosome markers like EEA1 and Lamp1 for confocal imaging. Unfortunately, we failed to obtain convincing results. One possible reason is that the titer of our virus stock is not high enough. The highest multiplicity of infection (MOI) we can use is less than 10. Using high MOI such as 50 as reported¹ might enhance the signal. Currently we still cannot concentrate the virions by ultracentrifugation in our BSL-3 facility. We apologize for this.

On the other hand, we know that Calu-3 expresses low level of CTSL and the entry of SARS-CoV-2 is dependent on TMPRSS2^{2, 3}; A549 cells have minimal TMPRSS2 expression⁴. Using TMPRSS2 or CTSL specific inhibitor could readily detect the differential entry pathways in different cell types. Also, in contrast to Sdel virus, genetic editing of CTSL in A549 cells has no impact on the infectivity of Sfull virus with intact spike protein. These functional results showed that the entry routes are cell type-dependent.

- L.89-91: How many independent passaging attempts were made in Vero - trypsin, Veros + trypsin and in Vero-TMPRSS2? Are the results shown in fig. S1 from just one experiment? Then the authors should be careful with interpretation. If they want to conclude that the presence of trypsin or TMPRSS2 impacts the selection of a deletion variant multiple passaging attempts need to be performed in parallel.

We appreciate the comment and agree. As shown in supplementary Fig. 1, we passaged the Sfull virus two times continuously (Sfull > P1 > P2) in Vero without trypsin, Vero with trypsin, or Vero expressing the TMPRSS2, and sequenced the two passages (P1 and P2). It was one independent continuous passaging experiment in each cell line. Currently, growing our virus stock in Vero E6 in the presence of trypsin has become a standard protocol. For the last 4 virus stocks prepared independently, we detected no deletion/mutation at the S1/S2 boundary site. However, we did a competition experiment and found that if the Sfull virus is mixed with a very small population of Sdel, like 1%, the Sdel will grow to a dominant population even in the presence of trypsin. This is consistent with our results as shown in Fig. 1b that Sdel infectivity is enhanced in both Vero + trypsin and Vero-TMPRSS2. To be cautious, we added “during a continuous passaging experiment” (page 4 line 137).

- Fig. 4A: In the results section it is not described how virus binding and internalization were measured.

We appreciate the suggestion. We have the detailed description in the “Methods”, and a brief explanation in the figure legend.

- Fig. 5: If the virus replicates 1-2 logs less it is not surprising that transmission does not work. So to me it is not clear what I am learning from the hamster data and what they add to the manuscript.

We appreciate the comment and agree with the Reviewer that decreased virus replication may affect the interpretation of transmission study. From our results, in the turbinate, the day 1 average titers of Sfull and Sdel reached 7.7 and 6.2 logs, respectively, indicating that both Sfull and Sdel viruses replicate efficiently. Moreover, at

day 2 and day 4 post-challenging, the difference in turbinate titers between Sfull and Sdel were not statically significant. After 3 days of co-housing (from day 1 to day 4), the turbinate tissue titer of Sfull-contacted hamsters reached 6.6 logs, but the titer of Sdel was under the limit of detection. It implied that the transmission of Sdel to the directly contacted naïve hamsters is inefficient or delayed as compared to the Sfull. In general, the animal data showed that the S1/S2 cleavage site modulates the infectivity and transmissibility, presumably due to usage of the more efficient plasma membrane fusion entry pathway by intact spike protein.

- Ref. 17 Something went wrong with reference 17. No paper is listed.
We have corrected this error.

Reference:

1. Daly JL, *et al.* Neuropilin-1 is a host factor for SARS-CoV-2 infection. *Science* **370**, 861-865 (2020).
2. Park JE, *et al.* Proteolytic processing of Middle East respiratory syndrome coronavirus spikes expands virus tropism. *Proceedings of the National Academy of Sciences of the United States of America* **113**, 12262-12267 (2016).
3. Hoffmann M, *et al.* SARS-CoV-2 Cell Entry Depends on ACE2 and TMPRSS2 and Is Blocked by a Clinically Proven Protease Inhibitor. *Cell* **181**, 271-280 e278 (2020).
4. Matsuyama S, *et al.* Enhanced isolation of SARS-CoV-2 by TMPRSS2-expressing cells. *Proceedings of the National Academy of Sciences of the United States of America* **117**, 7001-7003 (2020).

Reviewers' Comments:

Reviewer #3:

Remarks to the Author:

The authors addressed my comments well. In my opinion, this revised manuscript will be a valuable addition to the field.